# Clinical and Behavioural Heterogeneity Among Women at Increased Risk for Gestational Diabetes: A Four-Country Analysis

**DOI:** 10.3390/ijerph22071022

**Published:** 2025-06-27

**Authors:** Sharleen L. O’Reilly, Ellen Greene, Fionnuala M. McAuliffe, Helena Teede, Cristina Campoy, Christy Burden, Aisling Geraghty, Mercedes G. Bermúdez, Anna Davies, Cheryce L. Harrison, Helle Terkildsen Maindal, Vincent L. Versace, Ditte Hjorth Laursen, Timothy Skinner

**Affiliations:** 1School of Agriculture and Food Science, University College Dublin, D04 V1W8 Dublin, Ireland; ellen.greene@ucd.ie (E.G.);; 2UCD Perinatal Research Centre, School of Medicine, University College Dublin and National Maternity Hospital, D02 YH21 Dublin, Ireland; fionnuala.mcauliffe@ucd.ie; 3Monash Centre for Health Research and Implementation, School of Public Health and Preventive Medicine, Monash University, Melbourne 3800, Australia; helena.teede@monash.edu (H.T.); cheryce.harrison@monash.edu (C.L.H.); 4Department of Pediatrics, School of Medicine, University of Granada, 18012 Granada, Spain; ccampoy@ugr.es (C.C.); mgbermudez@ugr.es (M.G.B.); 5Instituto de Investigación Biosanitaria ibs.GRANADA, Health Sciences Technological Park, 18012 Granada, Spain; 6Academic Women’s Health Unit, Bristol Medical School, University of Bristol, Bristol BS8 1QU, UK; christy.burden@bristol.ac.uk (C.B.); anna.davies@bristol.ac.uk (A.D.); 7Department of Public Health, Aarhus University, 8000 Aarhus, Denmark; htm@ph.au.dk; 8Deakin Rural Health, Faculty of Health, Deakin University, Geelong 3220, Australia; vincent.versace@deakin.edu.au; 9GeoHealth Laboratory, Dasman Diabetes Institute, Kuwait City 15462, Kuwait; 10Liva Healthcare, 1434 Copenhagen, Denmark; 11School of Psychology, Faculty of Health, Deakin University, Geelong 3200, Australia; t.skinner@deakin.edu.au

**Keywords:** gestational diabetes mellitus, pregnancy, obesity, prevention, risk factors, health literacy

## Abstract

Gestational diabetes mellitus (GDM) is a growing global health concern due to its impact on maternal and infant health. GDM risk factors vary across populations, but international comparisons using standardised assessment tools are lacking. This study aimed to examine variations in risk factors, demographics and health behaviours among pregnant women at increased risk of GDM across four international sites and to investigate factors associated with maternal body mass index (BMI), a modifiable risk factor for GDM. This cross-sectional study included data from 804 pregnant women in Dublin (*n* = 213), Bristol (*n* = 205), Granada (*n* = 211) and Melbourne (*n* = 175) identified as having an increased risk of GDM, using the Monash GDM screening tool. Between-site differences were analysed using analysis of variance, Kruskal–Wallis and chi-square tests and factors associated with BMI at each site were examined using multiple linear regression. Despite standardised risk screening, significant heterogeneity was observed between sites in key GDM risk factors, including age (mean range 33.8–36.7 years), BMI (Melbourne 28.9 vs. Granada 26.9 kg/m^2^), physical activity (34.86–41.77 METs/week) and dietary intake (mean energy 1881–2136 kcal/day). Multiple factors were independently associated with BMI, including education level, ethnicity, health literacy and energy intake, with patterns varying by site. This study challenges the concept of a homogeneous “high-risk” GDM population by revealing substantial variations in risk factors and characteristics across different patient cohorts, highlighting the importance of developing context-sensitive approaches to GDM prevention.

## 1. Introduction

Gestational diabetes mellitus (GDM) is a growing health concern affecting up to 18% of pregnancies globally [1]. The prevalence of GDM varies significantly across populations, influenced by factors such as ethnicity, age, body mass index (BMI) and family history of diabetes [2,3]. GDM is associated with an increased risk of several adverse maternal (induction of labour, Caesarean delivery, large-for-gestational-age offspring, preterm delivery, congenital anomaly, hypertension, perinatal death, preeclampsia and shoulder dystocia [4,5,6,7,8,9]) and offspring outcomes, including type 2 diabetes development and obesity in children [10,11]. Given these risks, understanding the characteristics of women at risk of GDM is crucial for developing effective prevention strategies, particularly those relating to modifiable risk factors, such as body weight, diet and physical activity. Previous research has shown that health behaviour change interventions (diet and physical activity) can be successful in limiting excess gestational weight gain and reducing future chronic disease risk [12]. However, the effectiveness of such interventions may vary across different populations due to diverse cultural, socioeconomic and healthcare contexts, highlighting the importance of developing tailored strategies [13].

While GDM prevalence varies across populations [1], most risk assessment and prevention strategies assume relative homogeneity in how risk factors manifest across different cultural and healthcare contexts [14]. This assumption persists despite evidence that factors such as BMI carry different predictive weights across ethnic groups and healthcare settings [15]. The lack of international comparisons using standardised risk assessment tools has created a critical knowledge gap, where the level of heterogeneity in the characteristics of ‘high-risk’ cohorts across different populations is unclear. Understanding population heterogeneity is crucial for several reasons. Firstly, it informs whether current risk assessment tools, typically validated in limited cohorts, can be effectively applied to different populations settings. Secondly, it helps explain why some GDM prevention interventions show varying effectiveness across populations. Thirdly, it provides essential insights for adapting prevention strategies to local contexts. This is particularly important given that health literacy and healthcare engagement patterns, which can influence the success of interventions [16], often vary substantially across populations [17,18].

This study provides the first systematic international comparison of GDM risk profiles using a standardised risk assessment tool [3] across four countries with different healthcare systems and population demographics. By leveraging data from the Bump2Baby and Me multi-centre randomised trial, conducted across sites in Ireland, the UK, Spain and Australia, we address these knowledge gaps by examining (1) variations in GDM risk factors and clinical and demographic characteristics among women identified as having an increased risk of GDM across four countries; (2) population patterns in health and lifestyle behaviours; and (3) factors associated with BMI, a modifiable risk factor for GDM [3], in this high-risk population. These findings will contribute to a better understanding of how GDM risks present in diverse populations and potentially identify targeted areas for prevention in different healthcare settings.

## 2. Materials and Methods

### 2.1. Study Design and Participants

This cross-sectional study utilised baseline data from the Bump2Baby and Me (B2B&Me) multi-centre randomised controlled trial (RCT) (Australian New Zealand Clinical Trials Registry: ACTRN12620001240932) [19]. A power calculation was conducted to determine the sample size for the RCT, based on the primary outcome of maternal weight status at 12 months postpartum. The recruitment of 800 women was required to detect a 0.8 kg/m^2^ BMI difference between groups with a standard deviation of 2 kg/m^2^ BMI, with a type I error rate of 5%, and to achieve 80% power [19]. No separate power analysis was conducted for this cross-sectional baseline analysis, as the study utilises the full available baseline dataset to descriptively examine participant heterogeneity across sites. The study was conducted across four maternity services sites: National Maternity Hospital, Dublin (Ireland); North Bristol NHS Trust, Bristol (UK); Clinical University Hospital San Cecilio, Granada (Spain); and Monash Medical Centre, Melbourne (Australia). Ethical approvals were obtained from the National Maternity Hospital Human Research and Ethics Committee (EC18.2020), University College Dublin HREC-Sciences (LS-E-20-150-OReilly), NHS Health Research Authority and Health and Care Research Wales (21/WA/0022), Junta de Andalucía CEIM/CEI Provincial de Granada (2087-M1-22) and Monash Health HREC (RES-20-0000-892A). All participants provided informed consent prior to study enrolment.

Eligibility criteria included attending one of the participating maternity services, scoring ≥3 on the Monash GDM Screening tool [3], owning a smartphone capable of hosting the intervention app, sufficient understanding of the local language (English or Spanish) to provide informed consent and having a gestational age <24 weeks. Exclusion criteria were established diabetes, active cancer, severe mental illness, heart attack, substance abuse in the prior three months and multiple pregnancy.

### 2.2. Recruitment and Data Collection

Participants were recruited between February 2021 and April 2022. Women were screened for GDM risk at their first antenatal visit (typically 10–16 weeks’ gestation) by trained midwives and researchers. GDM risk scores were calculated using data on participant BMI, age, ethnicity, past history of GDM and family history of type 2 diabetes [3,19]. The standard practice for first antenatal visits at each site is detailed in Table 1. Due to the COVID-19 pandemic, recruitment procedures were adapted, varying slightly between sites. In general, eligible women were contacted (via phone or in person), provided with study information and invited to participate. In Dublin and Bristol, women were contacted by phone 1–7 days before their antenatal visit. In Granada, women were approached after their first antenatal visit. In Melbourne, women were contacted by phone after receiving study information via email or post. Consent and baseline data collection were conducted in person where possible or remotely when necessary, due to COVID-19 restrictions.

### 2.3. Measures

Participant characteristics were collected via case report forms and included maternal demographics (age, ethnicity and parity), socioeconomic factors (education, employment and housing) and medical history (previous GDM, family history of diabetes and smoking status). Height, weight and blood pressure were measured by trained researchers when possible; self-reported measurements were used when in-person visits were not feasible. BMI was calculated and categorised according to standard cut-offs (underweight (<18.5 kg/m^2^), normal weight (18.5–24.99 kg/m^2^), overweight (25.0–29.99 kg/m^2^) and obese (≥30.0 kg/m^2^)).

Participants completed online questionnaires including the full versions of the Food4Me food frequency questionnaire [20,21], Pregnancy Physical Activity Questionnaire (PPAQ) [22], EuroQol-5D-5L [23], Edinburgh Postnatal Depression Scale [24] and Pittsburgh Sleep Quality Index (PSQI) [25]. The Health Literacy Questionnaire (HLQ) [26] and eHealth Literacy Questionnaire (eHLQ) [27] used domains 2, 3, 6, 7 and 9 for the HLQ and 1, 4, 5 for eHLQ; average scores for each domain were calculated [26,27]. The EQ-5D visual analogue scale was used as a measure of perceived health, rated by participants on a scale from 0 to 100 [23]. A global sleep quality score, from 0 to 21, was generated through calculating the sum of the seven PSQI component scores [25]. Physical activity levels were assessed using the PPAQ scoring protocol [22]. Energy intake misreporting was evaluated using the Goldberg method [28] with coefficients of variation proposed by Black [29]. Age-specific Schofield equations were used to calculate basal metabolic rate (BMR).

### 2.4. Data Analysis

Statistical analyses were performed using IBM SPSS v27.0. Continuous variables were analysed using ANOVA when assumptions were met. Welch’s F test was used when variances were unequal between groups, as this method does not assume equal variances. Post hoc comparisons used Tukey’s HSD test after ANOVA and the Games–Howell test after Welch’s F test. Chi-square tests analysed categorical variables, with z-tests used for post hoc analysis. Kruskal–Wallis tests analysed ordinal data, with Dunn’s post hoc tests. ANCOVA assessed dietary variables, adjusting for energy intake, and Bonferroni post hoc tests were conducted. Prior to ANOVA and ANCOVA, normality was assessed using Kolmogorov–Smirnov tests and homogeneity of variances using Levene’s tests (Appendix A). ANOVA and ANCOVA were employed following the standard practice in epidemiological research, with this approach justified by adequate sample sizes and the clinical relevance of mean difference comparisons [30].

Categorical variables with low counts were collapsed, where appropriate, to improve the reliability of the analyses. Employment status was collapsed into ‘homemaker’, ‘full-time employment’, ‘part-time employment’ and ‘student, casual employment, government or disability supports’. Living circumstances were collapsed into ‘own home’, ‘rented home’ and ‘living with family or in emergency accommodation’. Ethnicity was dichotomised into ‘Caucasian’ and ‘all other ethnic groups’ for the analyses, as almost three quarters of participants identified as Caucasian and the remaining six ethnicity categories contained insufficient sample sizes to permit meaningful statistical analysis. Factors associated with maternal BMI for each site were explored using bivariate analysis. Spearman’s Rho correlations were conducted for continuous (age, gravida, parity, energy (Kcal), protein, carbohydrate and fat intake, physical activity levels and EQ-5D visual analogue scale) and ordinal (depression score, sleep score, eHealth literacy scores and health literacy scores) variables, while ANOVA and *t*-tests were used for categorical variables (study site; education level: ‘primary or secondary school’, ‘vocational qualification’, ‘university undergraduate degree’ or ‘university postgraduate degree’; ethnicity: ‘Caucasian’ or ‘other ethnic group’; living situation: ‘own home’, ‘rented home’ or ‘living with family or emergency accommodation’; employment: ‘full-time’, ‘part-time’, ‘homemaker’ or ‘student, casual employment, disability or government support’; marital status: ‘single, divorced or widowed’ or ‘married or de facto partner’; smoking now: ‘yes’ or ‘no’; smoked ever: ‘yes’ or ‘no’; and childcare responsibility: ‘yes’ or ‘no’). Variables significantly associated with BMI for each site (*p* < 0.05) were entered into separate multiple linear regression models. To avoid multicollinearity, variables with a variance inflation factor over 5 and that were highly correlated with each other were not entered into the model (r > 0.7); as energy intake was correlated with the other dietary variables, energy intake only was entered into the models.

For all analyses including dietary variables, implausible energy reporters were excluded. Plausible reporters were defined based on the ratio of reported energy intake to estimated BMR, using cut-offs derived from the Goldberg equation [28]. The potential impact of gestational age at baseline was explored using Spearman’s Rho correlations for diet and physical activity, behaviours where differences could impact outcomes of interest. Sedentary, household and total activity had significant associations with gestation (positive for sedentary and negative for total and household energy). ANCOVA was performed to test for differences in sedentary, household and total activity between sites, adjusting for gestation. However, the *p*-values produced from these tests did not differ from unadjusted *p*-values, so results from ANCOVA are not reported.

## 3. Results

### 3.1. Participant Characteristics

A total of 804 women were recruited across four sites (Figure 1), with participant numbers detailed in Table 2. Table 2 presents the participant characteristics of this high-risk cohort. Between-site differences in GDM risk factors showed varying participant risk profiles across countries. The mean age of participants was 35.2 years (SD 4.3), with differences between sites (F = 17.9; df = 3; *p* < 0.001). Post hoc analysis revealed significant age differences between sites (*p* < 0.001), with the youngest participants in Melbourne and the oldest in Granada. The mean BMI of the cohort was 27.7 kg/m^2^ (SD 6.6), with site differences also observed (Welch’s F = 3.6; df = 3; *p* = 0.014). Post hoc analysis showed that women in Melbourne had a higher mean BMI than those in Granada (28.9 kg/m^2^ vs 26.9 kg/m^2^, *p* = 0.024). BMI categories were distributed as follows: 0.1% underweight, 42.7% normal weight, 28.5% overweight and 28.7% obese.

The total cohort identified as being part of the following ethnic groups: 73.1% Caucasian, 3.9% Black or African American, 0.4% Indigenous Australian or Travelling, Roma or Gypsy Communities, 1.7% Arabic or Middle Eastern, 15.4% Asian, South Asian or Pacific Islander, 1.2% South East Asian and 4.2% Hispanic or Latino. Ethnicity differed between sites (χ^2^ = 147.4; df = 3; *p* < 0.001), where Melbourne had a significantly lower proportion of Caucasian participants (39.4%), with 46.4% of the Melbourne cohort identifying as Asian, South Asian or Pacific Islander. Granada had more Caucasian participants than other sites (92.9%). Previous history of GDM also varied between sites (χ^2^ = 26.5; df = 3; *p* < 0.001), with Melbourne having the highest proportion (14.3%) compared to Dublin (2.3%), Bristol (4.4%) and Granada (5.2%). There were no differences in the proportion of women with a family history of diabetes. Thus, GDM risk scores differed between sites, with Melbourne participants scoring higher than participants recruited from all other sites (H = 49.7; df = 3; *p* < 0.001).

On the other hand, mean gestational age at baseline was 13.9 weeks, but Melbourne participants completed baseline measures approximately 5 weeks later than women at other sites (Welch’s F = 264.8; df = 3; *p* < 0.001). No significant differences were seen in gravidity, parity or stillbirth across sites (Table 2).

Regarding sociodemographic variables, underlying differences were also observed. Significant differences were observed in education level (χ^2^ = 109.2; df = 9; *p* < 0.001): most participants (72.5%) had completed higher education, with the highest proportion in Dublin (79.2%) and the lowest in Bristol (71.7%). Significant differences were also observed in employment status, with approximately two thirds of participants being full-time employed (χ^2^ = 64.7; df = 9; *p* < 0.001), and living situation, (χ^2^ = 17.6; df = 6; *p* = 0.007). In total, 91% were married or in a de facto partnership, and women from the Granada site had a higher proportion of being single/divorced than those from other sites (*p* < 0.001). Slight differences were detected in current smoking habits between sites, and a significant difference in past smoking habits was also identified (*p* < 0.001), where a greater proportion of participants in Granada previously smoked in comparison to other sites (Table 2).

### 3.2. Blood Pressure

Blood pressure measurements differed between sites for both systolic (F = 9.1; df = 3; *p* < 0.001) and diastolic (Welch’s F = 10.5; df = 3; *p* < 0.001) readings. Post hoc analyses revealed that Bristol had significantly lower blood pressure readings compared to all other sites.

### 3.3. Health Behaviours

Sites differed for several key health behaviours (Table 3). Total energy (F = 7.1; df = 3; *p* < 0.001), protein (F = 15.1; df = 3; *p* < 0.001) and carbohydrate intakes (F = 6.6; df = 3; *p* < 0.001) were significantly different between sites. Dublin participants reported higher energy intake compared to Granada (*p* < 0.001) and Melbourne (*p* = 0.005). Bristol had a higher energy intake than Granada (*p* = 0.034).

Physical activity levels showed differences between sites in total activity (F = 6.6; df = 3; *p* < 0.001), household activities (including caregiving activities and household tasks) (F = 10.4; df = 3; *p* < 0.001) and sedentary activities (F = 10.6; df = 3; *p* < 0.001). Granada participants reported performing more household activities and being less sedentary compared to all other sites (*p* < 0.001 for all comparisons). The EQ-5D visual analogue scale, as a measure of health status, did not differ across sites.

### 3.4. Mental Health and Sleep Quality

The Edinburgh Postnatal Depression Scale revealed that 33.7% (*n* = 241) of participants had elevated depressive symptomatology, with no differences between sites (*p* = 0.641). Sleep quality was poor (score ≥ 5) in 72% (*n* = 508) of the 706 respondents, with Granada participants reporting poorer sleep quality than other sites (*p* < 0.001).

### 3.5. Health Literacy

Significant differences were observed between sites for all three digital eHealth literacy scales. Melbourne participants demonstrated superior digital health literacy across most domains, while Granada participants showed the highest confidence in digital data safety.

Melbourne respondents generally reported higher capabilities across health literacy domains, particularly in healthcare system navigation and engagement with providers. Dublin participants showed strengths in provider engagement, while Bristol participants demonstrated a better understanding of health information compared to Granada and Dublin cohorts.

### 3.6. BMI Associations

#### 3.6.1. Bivariate Analyses

In bivariate analysis, education level was the only factor consistently associated with BMI across all sites, with participants of higher education levels having lower BMIs. Health literacy, specifically the ‘perceived ability to actively manage health’ domain, was negatively correlated with BMI at three sites (Bristol, Granada and Melbourne). Other factors associated with BMI showed considerable variation between sites, highlighting population-specific influences. Age (r = −0.266; *p* < 0.001), marital status (t = 2.366; df = 184; *p* = 0.019) and childcare responsibility (t = 2.876; df = 113.847; *p* = 0.005) were associated with BMI in Bristol only. Participants’ living circumstances were associated with BMI in the Bristol (F = 5.404; df = 2; *p* = 0.005) and Granada (Welch’s F = 5.758; df = 2; *p* = 0.014) cohorts; in Bristol, those living in a rented home had a higher BMI than those who owned their home (mean difference = 3.618; *p* = 0.004), and in Granada, those living with family had a higher BMI than those who owned their home (mean difference = 9.346; *p* = 0.044). Gravida (r = 0.166; *p* = 0.029) and parity (r = 0.207; *p* = 0.006) were positively associated with BMI in the Melbourne cohort, and ethnicity was also associated with BMI (t = −3.591; df = 115.196; *p* < 0.001), where Caucasian participants had a higher BMI than other ethnicities. In the Granada cohort, employment status was associated with BMI (F = 3.167; df = 3; *p* = 0.026); those in the ‘student, casual employment, government or disability support’ group had a higher mean BMI than those in full-time employment (mean difference = 3.285; *p* = 0.029). In Granada, those smoking in early pregnancy had a higher BMI than those who did not smoke (t = 2.228; df = 200; *p* = 0.027).

Total physical activity level was positively correlated with BMI (r = 0.170; *p* = 0.017) in the Granada cohort, and engaging in sports was also positively correlated with BMI for the Dublin cohort (r = 0.155; *p* = 0.036). Self-reported health status (EQ-5D visual analogue scale) was negatively correlated with BMI in the Dublin (r = −0.169; *p* = 0.020) and Melbourne (r = −0.232; *p* = 0.003) cohorts. Aspects of health literacy were correlated with BMI in all sites; in Dublin, the scale measuring participants’ perceived ability to actively engage with healthcare providers was negatively correlated with BMI (r = −0.165; *p* = 0.025), and in Bristol (r = −0.236; *p* = 0.002), Granada (r = −0.234; *p* = 0.001) and Melbourne (r = −0.190; *p* = 0.017), the scale measuring the perceived ability to actively manage health was negatively correlated with BMI.

#### 3.6.2. Multivariate Analyses

Across the multivariate models, several patterns emerged consistently. Education level remained a significant predictor of BMI in Bristol and showed trends in other sites. Energy intake was positively associated with BMI in both sites when dietary data were included (Bristol and Melbourne). Health literacy, particularly the ability to actively manage one’s health, was negatively associated with BMI in both Bristol and Granada models.

##### Dublin

None of the variables in the regression model for the Dublin cohort were associated with BMI (Table 4).

##### Bristol

In the Bristol cohort, age (B = −0.291; 95% CI = [−0.553, −0.030;] *p* = 0.029), the health literacy scale measuring the perceived ability to actively manage health (B = −2.426; 95% CI = [−4.333, −0.519]; *p* = 0.013) and energy intake (B = 0.004; 95% CI = [0.002, 0.006]; *p* < 0.001) were associated with BMI. Education was also associated with BMI in the Bristol cohort, where those with undergraduate (B = −3.432; 95% CI = −5.897, −0.966; *p* = 0.007) and postgraduate degrees (B = −3.004; 95% CI = [−5.957, −0.051]; *p* = 0.046) had lower BMIs than those with primary or secondary school education.

##### Granada

For the model examining the influences on BMI in the Granada cohort, the “student, casual, government or disability support” group had higher BMIs than those in full-time employment (B = 2.719; 95% CI = [0.580, 4.859]; *p* = 0.013). Those living with family or in emergency accommodation had higher BMIs than those who owned their homes (B = 9.844; 95% CI = [5.564, 14.125]; *p* < 0.001). The health literacy score for actively managing health was negatively associated with BMI (B = −2.339; 95% CI = [−3.616, −1.061]; *p* < 0.001).

##### Melbourne

In the Melbourne cohort, energy was positively associated with BMI (B = 0.004; 95% CI = [0.001, 0.007]; *p* = 0.022). Ethnicity was associated with BMI, where women from non-Caucasian ethnic groups had lower BMIs than Caucasian participants (B = −4.494; 95% CI = [−7.251, −1.736]; *p* = 0.002).

## 4. Discussion

This international study revealed substantial heterogeneity in GDM risk factors and characteristics among women identified as high-risk for GDM, using a standardised screening tool. We found significant variations in clinical risk factors, health behaviours and health literacy across sites, challenging the notion of a uniform “high-risk” profile for GDM. Our analysis revealed that age, indicators of socioeconomic status, diet, ethnicity and the perceived ability to manage health were significantly associated with BMI in this study population, with differences observed between countries. Overall, we have identified population heterogeneity in GDM risk factors spanning demographic, clinical, and lifestyle domains.

The observed BMI patterns revealed significant regional differences, with Melbourne participants recording substantially higher BMIs than their Granada counterparts. This variation aligns with national obesity trends, as Australia reports nearly double the female adult obesity rate (31%) compared to Spain (17%) [31,32]. However, the BMI disparity in our study population exceeds the expected difference suggested by national statistics. The interpretation of these BMI variations requires careful consideration, particularly in multiethnic populations such as Australia, as research indicates that BMI’s predictive value for GDM varies across ethnic groups [15]. This finding has direct implications for interventions targeting diverse populations, as standardised BMI-based approaches may inadequately identify risk in some ethnic groups whilst over-classifying others. This may mean that ethnicity-specific risk thresholds and tailored prevention strategies would be more successful in these populations. With BMI being a potential modifiable risk factor for GDM [3], our analysis provides valuable evidence supporting established relationships between BMI and factors such as ethnicity and education level [33], whilst revealing opportunities for intervention refinement. Although structured dietary and physical activity support remains the cornerstone for managing gestational weight gain [34], the modest variance explained by our model suggests substantial unexplored influences on BMI in this high-risk population. Our findings suggest that women with elevated BMIs may benefit from broader strategies to enhance overall health, wellbeing and self-efficacy in addition to dietary changes, with further research needed.

Our study cohort demonstrated consistently higher maternal ages compared to national averages, particularly in Granada and Dublin, likely due to age being a component of the risk screening tool applied to identify participants [3]. This finding also reflects the broader trend of delayed childbearing in high-income countries [35,36]. The Granada cohort’s mean age of 36.7 years significantly exceeded Spain’s national average age of 31.2 years for first-time mothers in 2020 [37], a particularly relevant finding given that each yearly increase in maternal age corresponds to a 3–4% higher GDM risk [38,39]. However, we also observed indicators of higher socioeconomic status in the study cohort, such as high educational attainment, full-time employment and stable living environments, which may be related to the older maternal age profile and recruitment bias in clinical trials [40]. We identified an inverse relationship between education levels and BMI, consistent with established research showing socioeconomic status as a key determinant of BMI in both pregnant and non-pregnant populations [41,42]. Our findings indicate that prevention strategies may be adapted to differing demographic profiles; for instance, in populations with older maternal age profiles, as seen in Granada, prevention strategies might need to particularly emphasise age-related risks while acknowledging the potential protective factors associated with higher education levels and stable living environments often seen in this demographic. For populations with higher proportions of previous GDM, as observed in Melbourne (reflecting greater detection through universal GDM screening and higher rates of maternal obesity), prevention strategies may focus more on inter-pregnancy weight management and early intervention.

These demographic and behavioural patterns, along with variations in digital and health literacy across sites, have important implications for developing and implementing effective GDM prevention strategies and communication approaches. They highlight potential ways for GDM prevention programmes to be adapted to align with lifestyle norms and emphasise that differing modes of intervention delivery may be required across different populations; for example, more accessible messaging may be required in populations with lower levels of health literacy, whilst culturally adapted dietary recommendations and physical activity approaches may be needed in multicultural populations like Melbourne, where ethnicity is significantly associated with BMI patterns. The consistency of education and health literacy associations across sites suggests that these represent more universal determinants of BMI, transcending cultural and healthcare system differences. These findings align with the established literature linking socioeconomic factors to health outcomes and suggest that interventions addressing health literacy and self-efficacy may have broad applicability across diverse populations for weight management.

Our study’s primary strength lay in its systematic approach to GDM risk assessment across four international sites, enabling a robust international comparison of risk profiles within different healthcare contexts. The comprehensive dataset captures maternal demographics, lifestyle and clinical factors from a large international cohort, providing valuable insights into variations between European and Australian populations. Furthermore, we successfully achieved our target sample size based on a priori power estimates, lending statistical rigour to our findings.

However, several limitations warrant consideration. While we used a standardised risk assessment tool [3], validated in multiethnic and diverse cohorts [3,43,44], it had not been yet tested across all participating countries nor compared to local screening standards. Our findings suggest that whilst the Monash tool successfully identified high-risk populations across diverse settings, the variations in risk profiles observed indicate that screening tools may benefit from population-specific calibration or weighting of risk factors. For instance, the higher GDM risk scores in Melbourne, despite similar individual risk factor prevalence, suggest that ethnic composition and previous GDM history may require enhanced weighting in certain populations. Our focus on high-income countries may also limit the broader applicability of our findings. Additionally, whilst we observed significant ethnic diversity, particularly in Melbourne, sample size constraints prevented the detailed examination of specific ethnic subgroups, limiting our ability to provide ethnicity-specific recommendations for consideration. The COVID-19 pandemic significantly impacted our study protocol, necessitating site-specific adaptations to recruitment and data collection procedures. These adaptations led to between-site differences in gestational age at baseline and may have influenced participant profiles. Additional methodological limitations include the cross-sectional design, which precludes causal inference, and the reliance on subjective physical activity assessment. The latter is particularly noteworthy given the established evidence that individuals with higher adiposity tend to over-report activity levels [45,46], consistent with the negative association between activity levels and BMI observed in our cohort. The reduction of ethnicity to a binary variable, due to differing levels of diversity in the cohorts studied, is a limitation in terms of generalizability across ethnicities. Future research conducted in different contexts, with a focus on culturally and linguistically diverse groups, should explore potential ethnic-specific effects. The observed variations in GDM history between sites likely reflect differences in national screening practices and ethnic composition, particularly in Melbourne, where higher-risk ethnic groups are more prevalent.

## 5. Conclusions

This international study provides compelling evidence that women at high risk of GDM represent a diverse population with distinct demographic, clinical and behavioural characteristics shaped by ecosocial drivers and their healthcare contexts. Despite the challenges posed by conducting research during the COVID-19 pandemic, our findings offer important insights for advancing the understanding of GDM risk and, potentially, prevention strategies. The marked variations in risk profiles across study sites underscore the importance of moving beyond one-size-fits-all approaches to prevention. Our results establish a robust foundation for developing targeted interventions that acknowledge and respond to this heterogeneity. Future research should prioritise the development and validation of adapted risk assessment tools, while longitudinal studies will be essential to understand how these baseline characteristics influence GDM development and pregnancy outcomes across different settings. Specifically, follow-up of this well-characterised international cohort will provide valuable insights into whether the observed risk factor heterogeneity translates into different patterns of GDM onset, gestational weight gain and maternal–infant outcomes, informing more precise risk prediction and intervention targeting. This work represents a significant step towards more sophisticated, context-sensitive approaches to GDM prevention that can better serve diverse populations of women at risk.

## Figures and Tables

**Figure 1 ijerph-22-01022-f001:**
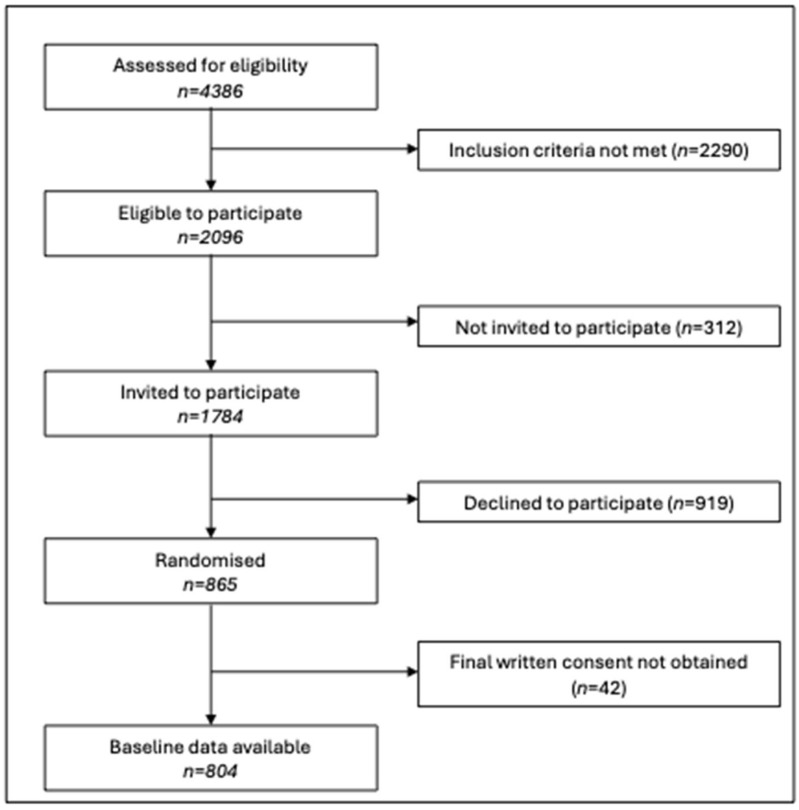
Participant recruitment and retention figures for the B2B&Me RCT.

**Table 1 ijerph-22-01022-t001:** Usual care for the first antenatal visit within the four participating clinical sites in Dublin, Bristol, Granada and Melbourne.

Process	Dublin	Bristol	Granada	Melbourne
Notification of pregnancy	Direct contact to hospital, administrative staff managed	Community midwife contacted	Community health centre contacted	Direct contact to hospital, administrative staff managed
First antenatal visit booking and risk screening	Administrative staff and research team conducted on phone	Community midwife, conducted on phone	Community-based, conducted locally. Risk screening performed at first antenatal visit	Administrative staff and research team conducted by email and/or phone
First antenatal visit	Midwife-led, in-person with venepuncture (non-fasting)	Dating scan visit, in person, venepuncture possible (non-fasting)	Midwife or clinician-led, in person, venepuncture possible (non-fasting)	Midwife or clinician-led via telehealth, venepuncture request sent for remote phlebotomy (non-fasting or fasting)

**Table 2 ijerph-22-01022-t002:** Participant characteristics at Dublin (*n* = 213), Bristol (*n* = 205), Granada (*n* = 211) and Melbourne (*n* = 175) sites.

Characteristics	Dublin	Bristol	Granada	Melbourne	Total	*p*
Age, mean (SD) (*n* = 804)	35.5 (4.1) ^a^	34.5 (4.2) ^ac^	36.7 (4.0) ^b^	33.8 (4.3) ^c^	35.2 (4.3)	**<0.001**
Ethnicity (*n* = 804)						**<0.001**
Caucasian	161 (75.6%) ^a^	162 (79.0%) ^a^	196 (92.9%) ^b^	69 (39.4%) ^c^	588 (73.1%)	
All other ethnic groups	52 (24.4%) ^a^	43 (21.0%) ^a^	15 (7.1%) ^b^	106 (60.6%) ^c^	216 (26.9%)	
**BMI** (kg/m^2^), mean (SD) (*n* = 804)	27.2 (5.9) ^abc^	28.2 (6.6) ^abc^	26.9 (5.5) ^b^	28.9 (8.2) ^c^	27.7 (6.6)	**0.014 ^d^**
BMI category (*n* = 804)						0.472
Underweight	0 (0.0%)	1 (0.5%)	0 (0.0%)	0 (0.0%)	1 (0.1%)	
Normal weight	97 (45.5%)	79 (38.5%)	96 (45.5%)	71 (40.6%)	343 (42.7%)	
Overweight	60 (28.2%)	62 (30.2%)	62 (29.4%)	45 (25.7%)	229 (28.5%)	
Obese	56 (26.3%)	63 (30.7%)	53 (25.1%)	59 (33.7%)	231 (28.7%)	
Previous gestational diabetes (*n* = 804)	5 (2.3%) ^a^	9 (4.4%) ^a^	11 (5.2%) ^a^	25 (14.3%) ^b^	50 (6.2%)	**<0.001**
Family history of diabetes (*n* = 804)	70 (32.9%)	64 (31.2%)	83 (39.3%)	64 (36.6%)	281 (35.0%)	0.302
GDM risk score, mode (minimum, maximum) (*n* = 804)	3 (3,7) ^a^	3 (3, 7) ^a^	3 (3, 6) ^a^	3 (3, 8) ^b^	3 (3, 8)	**<0.001**
Gestational age, mean (weeks) (*n* = 798)	12.8 (1.6) ^a^	12.9 (1.3) ^a^	13.0 (2.7) ^a^	17.9 (2.3) ^b^	14.0 (2.9)	**<0.001 ^d^**
Gravida (*n* = 797)						0.225
1	53 (25.4%)	54 (26.5%)	48 (22.7%)	46 (26.6%)	201 (25.2%)	
2	59 (28.2%)	82 (40.2%)	79 (37.4%)	49 (28.3%)	269 (33.8%)	
≥3	97 (46.4%)	68 (33.3%)	84 (39.8%)	78 (45.1%)	327 (41.0%)	
Parity (*n* = 797)						0.216
0	78 (37.3%)	82 (40.2%)	79 (37.4%)	64 (37.0%)	303 (38.0%)	
1	76 (36.4%)	97 (47.5%)	99 (46.9%)	76 (43.9%)	348 (43.7%)	
2	39 (18.7%)	16 (7.8%)	23 (10.9%)	25 (14.5%)	103 (12.9%)	
≥3	16 (7.7%)	9 (4.4%)	10 (4.7%)	8 (4.6%)	43 (5.4%)	
Previous stillbirth (*n* = 596)	3 (1.9%)	3 (2.0%)	3 (1.8%)	5 (3.9%)	14 (2.3%)	0.661 ^e^
Education (*n* = 743)						**<0.001**
Primary/secondary school	8 (4.2%) ^a^	47 (25.5%) ^b^	23 (11.4%) ^c^	32 (19.4%) ^b^	110 (14.8%)	
Vocational qualifications	32 (16.7%) ^a^	5 (2.7%) ^b^	52 (25.7%) ^c^	5 (3.0%) ^b^	94 (12.7%)	
University degree	65 (33.9%) ^a^	85 (46.2%) ^b^	74 (36.6%) ^ab^	74 (44.8%) ^b^	298 (40.1%)	
Postgraduate degree	87 (45.3%) ^a^	47 (25.5%) ^b^	53 (26.2%)^b^	54 (32.7%) ^b^	241 (32.4%)	
Employment (*n* = 738)						**<0.001**
Homemaker	18 (9.3%) ^ab^	8 (4.4%) ^b^	18 (9.0%) ^ab^	21 (12.8%) ^a^	65 (8.8%)	
Government assistance/disability support, student or casual employment	7 (3.6%) ^a^	6 (3.3%) ^a^	24 (12.1%) ^b^	16 (9.8%) ^b^	53 (7.2%)	
Part-time employment	17 (8.8%) ^a^	56 (30.8%) ^b^	28 (14.1%) ^a^	45 (27.4%) ^b^	146 (19.8%)	
Full-time employment	151 (78.2%) ^a^	112 (61.5%) ^b^	129 (64.8%) ^b^	82 (50.0%) ^c^	474 (64.2%)	
Marital status (*n* = 745)						**<0.001**
Single/divorced/widowed	11 (5.7%) ^a^	9 (4.8%) ^a^	42 (20.8%) ^b^	4 (2.4%) ^a^	66 (8.9%)	
Married or de facto partner	181 (94.3%) ^a^	177 (95.2%) ^a^	160 (79.2%) ^b^	161 (97.6%) ^a^	679 (91.1%)	
Living situation (*n* = 743)						**0.007**
Own home	117 (60.3%) ^a^	132 (71.4%) ^b^	151 (75.5%) ^b^	96 (58.5%) ^a^	496 (66.8%)	
Rented home	65 (33.5%) ^a^	45 (24.3%) ^b^	42 (21.0%) ^b^	56 (34.1%) ^a^	208 (28.0%)	
Living with family or in emergency accommodation	12 (6.2%) ^a^	8 (4.3%) ^a^	7 (3.5%) ^a^	12 (7.3%) ^a^	39 (5.2%)	
Children to care for throughout workday (*n* = 748)						**0.006**
No	65 (33.3%) ^a^	58 (31.2%) ^ab^	41 (20.3%) ^c^	39 (23.6%) ^bc^	203 (27.1%)	
Yes	55 (28.2%) ^a^	67 (36.0%) ^ab^	87 (43.1%) ^b^	74 (44.8%) ^b^	283 (37.8%)	
No children to care for at present	75 (38.5%) ^a^	61 (32.8%) ^a^	74 (36.6%) ^a^	52 (31.5%) ^a^	262 (35.0%)	
Current smoker (*n* = 748)	5 (2.6%) ^a^	10 (5.4%) ^ab^	14 (6.9%) ^b^	3 (1.8%) ^a^	32 (4.3%)	**0.049**
Previous smoker (*n* = 724)	55 (28.9%) ^a^	59 (32.4%) ^a^	84 (44.2%) ^b^	39 (24.1%) ^a^	237 (32.7%)	**<0.001**
BP systolic, mean (SD) (*n* = 631)	114.6 (11.0) ^a^	108.4 (11.5) ^b^	114.6 (11.4) ^a^	112.8 (14.5) ^a^	113.0 (12.1)	**<0.001**
BP diastolic, mean (SD) (*n* = 631)	70.6 (8.0) ^a^	65.8 (8.8) ^b^	70.8 (8.4) ^a^	70.7 (11.2) ^a^	69.7 (9.1)	**<0.001 ^d^**

Bold text denotes a statistically significant finding in table. GDM: gestational diabetes mellitus; BP: blood pressure; SD: standard deviation; ^a–c^: significant differences between sites; ^d^: violated assumption of homogeneity of variances so *p* value from Welch’s test used and the Games–Howell post hoc test reported. ^e^: *p* value from the Fisher–Freeman–Halton exact test reported.

**Table 3 ijerph-22-01022-t003:** Lifestyle and health behaviour variance across in Dublin (*n* = 213), Bristol (*n* = 205), Granada (*n* = 211) and Melbourne (*n* = 175).

	Dublin	Bristol	Granada	Melbourne	Total	*p*
Dietary intake (*n* = 482)						
Energy (kcal/day)	2136.21 (508.90) ^a^	2048.51 (481.53) ^ab^	1881.88 (506.54) ^c^	1906.98 (498.34) ^bc^	1995.35 (508.51)	**<0.001**
Fat (g/day)						
Unadjusted mean (SD)	93.63 (26.57)	89.23 (24.52)	83.83 (25.46)	82.28 (26.83)	87.44 (26.07)	
Adjusted mean (SE)	87.25 (1.13)	86.82 (1.10)	88.97 (1.06)	86.28 (1.28)		0.355 ^d^
Protein (g/day)						
Unadjusted mean (SD)	87.42 (24.09)	80.14 (24.04)	86.07 (25.26)	77.88 (20.87)	83.26 (24.08)	
Adjusted mean (SE)	82.23 (1.38) ^a^	78.18 (1.35) ^a^	90.25 (1.30) ^b^	81.13 (1.57) ^a^		**<0.001 ^d^**
Carbohydrate (g/day)						
Unadjusted mean (SD)	243.59 (73.08)	238.70 (66.77)	202.05 (61.50)	222.85 (71.67)	226.32 (69.86)	
Adjusted mean (SE)	227.46 (3.35) ^a^	232.61 (3.27) ^a^	215.05 (3.14) ^b^	232.98 (3.81) ^a^		**<0.001 ^d,e^**
Activity (METs/week, *n* = 716)						
Total	35.40 (19.19) ^a^	35.24 (15.95) ^a^	41.77 (18.65) ^b^	34.86 (16.61) ^a^	36.99 (17.93)	**<0.001**
Household activities	15.04 (11.43) ^a^	14.77 (10.85) ^a^	20.31 (12.37) ^b^	14.95 (11.23) ^a^	16.40 (11.74)	**<0.001**
Occupational activities	8.22 (9.02)	7.86 (6.42)	8.64 (9.15)	8.09 (7.23)	8.21 (8.09)	0.815 ^f^
Sport/exercise activities	5.52 (5.71)	5.72 (4.94)	6.05 (6.49)	5.17 (4.13)	5.64 (5.45)	0.440 ^f^
Transportation activities	3.08 (2.90)	3.08 (2.84)	3.62 (2.67)	2.91 (3.13)	3.19 (2.88)	0.092
Sedentary activities	7.70 (3.11) ^a^	7.84 (2.98) ^a^	6.45 (3.23) ^b^	8.18 (3.34) ^a^	7.50 (3.23)	**<0.001**
EQ-5D visual analogue scale (*n* = 732)	82.41 (15.04)	80.85 (16.24)	83.90 (14.69)	83.37 (13.80)	82.63 (15.02)	0.218
EPDS (*n* = 717)	6.91 (4.30)	7.29 (4.80)	7.08 (4.31)	7.47 (4.38)	7.18 (4.45)	0.641
Pittsburgh sleep score (*n* = 706)	6.26 (3.07) ^a^	6.75 (3.25) ^a^	7.56 (3.48) ^b^	6.47 (3.15) ^a^	6.79 (3.28)	**<0.001**
Infant feeding intention (*n* = 712)						
Breastfeed	121 (66.1%) ^a^	116 (65.5%) ^a^	157 (80.1%) ^b^	100 (64.1%) ^a^	494 (69.4%)	**<0.001**
Formula	12 (6.6%) ^a^	4 (2.3%) ^b^	5 (2.6%) ^ab^	4 (2.6%) ^ab^	25 (3.5%)	
Mixed feeding	40 (21.9%) ^a^	39 (22.0%) ^a^	16 (8.2%) ^b^	42 (26.9%) ^a^	137 (19.2%)	
No plan	10 (5.5%) ^a^	18 (10.2%) ^a^	18 (9.2%) ^a^	10 (6.4%) ^a^	56 (7.9%)	
eHealth literacy scores (*n* = 713)						
Using technology to process health information	2.92 (0.44) ^a^	2.94 (0.54) ^a^	2.93 (0.52) ^a^	3.11 (0.51) ^b^	2.97 (0.50)	**0.003**
Motivated to engage with digital services	2.81 (0.49) ^ab^	2.76 (0.56) ^a^	2.77 (0.48) ^a^	2.93 (0.48) ^b^	2.81 (0.51)	**0.038**
Feel safe and in control	2.83 (0.46) ^a^	2.84 (0.52) ^ac^	3.06 (0.57) ^b^	2.96 (0.51) ^c^	2.93 (0.53)	**<0.001**
Health literacy scores (*n* = 712)						
Having sufficient information to manage health	3.04 (0.48)	2.97 (0.52)	2.97 (0.52)	3.00 (0.49)	2.99 (0.50)	0.468
Actively managing health	2.78 (0.49)	2.72 (0.50)	2.66 (0.53)	2.75 (0.47)	2.72 (0.50)	0.063
Ability to actively engage with healthcare providers	3.92 (0.66) ^ac^	3.73 (0.71) ^b^	3.84 (0.68) ^ab^	4.05 (0.58) ^c^	3.88 (0.67)	**<0.001**
Navigating the healthcare system	3.62 (0.62) ^a^	3.51 (0.69) ^a^	3.60 (0.64) ^a^	3.77 (0.70) ^b^	3.62 (0.66)	**0.011**
Understand health information	4.16 (0.50) ^a^	4.27 (0.56) ^b^	4.05 (0.55) ^a^	4.35 (0.51) ^b^	4.20 (0.54)	**<0.001**

Bold text denotes a statistically significant finding in table. EQ-5D: health-related quality of life questionnaire; EPDS: Edinburgh Postnatal Depression Score; MET: Metabolic Equivalent of Task; SD: standard deviation; SE: standard error; ^a–c^: significant differences between sites; ^d^: adjusted for energy intake; ^e^: violated assumption of homogeneity of variances; ^f^: violated assumption of homogeneity of variances. *p* value from Welch’s test is reported. Values are means (standard deviations), unless otherwise specified.

**Table 4 ijerph-22-01022-t004:** Multiple linear regression model to assess associations with maternal BMI at each site.

	Dublin (*n* = 177)		Bristol (*n* = 122)		Granada (*n* = 191)		Melbourne (*n* = 92)	
	B (95% CI)	VIF	B (95% CI)	VIF	B (95% CI)	VIF	B (95% CI)	VIF
Education level (reference category: primary or secondary school level)								
Vocational qualification	−0.162 (−4.791, 4.466)	4.505	7.322 (−3.012, 17.655)	1.068	−1.228 (−3.849, 1.393)	2.821	−1.720 (−9.668, 6.229)	1.628
Undergraduate education	−1.854 (−6.263, 2.555)	6.391	−3.432 (−5.897, −0.966) **	1.869	−2.569 (−5.148, 0.011)	3.509	−4.413 (−9.189, 0.363)	3.533
Postgraduate education	−3.631 (−7.986, 0.724)	6.807	−3.004 (−5.957, −0.051) *	1.988	−1.488 (−4.169, 1.192)	3.236	−3.675 (−8.846, 1.497)	3.813
Age			−0.291 (−0.553, −0.030) *	1.331				
Childcare responsibility								
Yes			0.754 (−1.319, 2.828)	1.219				
Living circumstances (reference category: own home)								
Rented home			0.423 (−1.944, 2.790)	1.155	1.083 (−0.634, 2.799)	1.076		
Living with family or in emergency accommodation			−2.599 (−8.801, 3.603)	1.135	9.844 (5.564, 14.125) ***	1.252		
Marital status								
Single			0.436 (−5.561, 6.433)	1.061				
Employment status (reference category: full-time employment)								
Homemaker					−1.007 (−3.522, 1.508)	1.152		
Student, casual employment, government or disability support					2.719 (0.580, 4.859) *	1.089		
Part-time employment					1.435 (−0.537, 3.407)	1.093		
Gravida							−0.361 (−1.300, 0.578)	1.697
Parity							0.493 (−1.453, 2.438)	1.675
Ethnicity (reference category: Caucasian)								
All other ethnic groups							−4.494 (−7.251, −1.736) **	1.133
Energy intake (Kcal)			0.004 (0.002, 0.006) ***	1.075			0.004 (0.001, 0.007) *	1.333
Physical activity (METs/week)								
Activity level from sport	0.137 (−0.017, 0.291)	1.001						
Total physical activity level					0.032 (−0.006, 0.069)	1.100		
Health literacy scale (average score)								
Ability to actively engage with healthcare providers	−0.599 (−1.887, 0.689)	1.073						
Actively managing my health			−2.426 (−4.333, −0.519) *	1.148	−2.339 (−3.616, −1.061) ***	1.047	−1.962 (−5.123, 1.198)	1.367
EQ-5D visual analogue scale	−0.049 (−0.109, 0.011)	1.069					−0.048 (−0.150, 0.055)	1.253
Current smoking (yes)					1.597 (−1.021, 4.215)	1.046		

CI: confidence interval; VIF: variance inflation factor; EQ-5D: health-related quality of life questionnaire; MET: Metabolic Equivalent of Task. * *p* < 0.05; ** *p* < 0.01; *** *p* < 0.001. Dublin: R^2^ = 0.100; model *p* = 0.006. Bristol: R^2^ = 0.331; model *p* < 0.001. Granada: R^2^ = 0.282; model *p* < 0.001. Melbourne: R^2^ = 0.293; model *p* < 0.001.

## Data Availability

The datasets generated during and/or analysed during the current study are available from the corresponding author on reasonable request, subject to approval from the relevant ethics committees and in accordance with the data protection regulations of the participating institutions.

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
