# Peer review of "Clinical and Behavioural Heterogeneity Among Women at Increased Risk for Gestational Diabetes: A Four-Country Analysis"

_ijerph, 2025, doi:10.3390/ijerph22071022_

Round 1
Reviewer 1 Report
Comments and Suggestions for Authors
- Title: "Clinical, Lifestyle, and Behavioral Heterogeneity Among Women at Increased Risk for Gestational Diabetes: A Four-Country Study" — well-formulated.
- Abstract: Acceptable.
- Introduction: Overall, well written.
- Methodology: Please provide a detailed explanation of how the assessment for the GDM risk score, eHealth literacy scores, eHealth, EQ-5D visual analogue scale, and Pittsburgh Sleep Quality Index was conducted.
- Results: The results are well explained. However, Tables 4–7 might be rearranged into one or two tables to improve clarity.
- References: Please update with more recent references, except for foundational older ones.
The English could be improved to help the reader understand better.
Author Response
Dear Reviewer,
We thank you for taking the time to review this manuscript. Please find the detailed responses attached and the corresponding revisions in red font in the re-submitted file.

Reviewer 2 Report
Comments and Suggestions for Authors
This manuscript is a cross-sectional study that compares maternal gestational diabetes mellitus (GDM) risk across international regions and analyzes factors associated with maternal body mass index (BMI) using a multiple linear regression model. It provides valuable findings on how health-related risk factors including education level, activity level, EQ5D by region worldwide.
Please refer and clarify below comments:
3. Results
It seems that the prerequisites for main statistical analysis, such as the normality test and the equal variance test, were considered in detail. However, if some tables of results for each variable for such prerequisites is presented, it will be recognized as a more reliable research result, so I would like to give authors some addition as follows;
1) In line 93-94, when presenting the “Ethnicity” variable in Table 2, it is recommended to consolidate all ethnicities other than Caucasian into a single category labeled “Others,” so that the cross-tabulation clearly compares Caucasian with other ethnicities.
2) In line 52, before presenting the ANOVA results for continuous variables in Tables 2 and 3, I recommend adding separate tables that report the Kolmogorov-Smirnov normality test and Levene’s test for homogeneity of variances for those continuous variables.
3) In line 199, It is recommended that the multicollinearity diagnostics for each variable in Tables 4 and 5 be reported as variance inflation factors (VIFs).
Comments on the Quality of English Language
A minor revision of the English language may be necessary.
Author Response

(The authors gave the same response as above.)

Reviewer 3 Report
Comments and Suggestions for Authors
Dear authors...
Thank you for this article...
Here is my review...
1. General Assessment
This manuscript presents a well-executed, cross-national study exploring clinical, sociodemographic, and behavioural heterogeneity among pregnant women at increased risk for gestational diabetes mellitus (GDM) across four high-income countries: Ireland, the UK, Spain, and Australia. The use of a standardized risk screening tool (Monash GDM Screening Tool) across sites allows for meaningful international comparison, addressing a well-recognized gap in GDM literature.
The study is timely and relevant, especially in the context of global efforts to refine early prevention and risk stratification strategies for GDM. The data presented are rich, the analyses are appropriate, and the findings largely support the conclusions.
2. Abstract
The abstract is well structured, effectively summarizing the aim, methodology, key findings, and conclusions. However, it lacks specific quantitative information that would make the results more impactful.
Suggestions:
-
Include specific site-level differences in BMI, physical activity, or health literacy.
-
Mention the number of participants recruited from each country
3. Introduction
The introduction offers a thorough background on GDM, its implications, and why a nuanced, context-sensitive understanding of high-risk groups is needed. The rationale for international comparison and the use of a standard screening tool is well justified.
Strengths:
-
Logical flow and clear identification of the knowledge gap.
-
Solid grounding in literature regarding GDM risk variability across populations.
Minor Improvement:
-
Clarify the contribution of the current study about existing international comparative efforts in GDM.
4. Methods
The study design is clearly described, and ethical approvals are appropriately reported. Data sources, recruitment procedures, measurement instruments, and statistical analyses are rigorously documented.
Strengths:
-
Use of validated tools (e.g., PPAQ, Food4Me, EPDS, HLQ).
-
Clear statistical strategy, including handling of implausible energy reporters.
Limitations:
-
The gestational age at baseline differed significantly between sites (notably in Melbourne), which could influence behavioural data.
-
The reliance on self-reported anthropometric and behavioural data due to COVID-19 restrictions may introduce bias.
Suggestions:
-
Discuss the potential impact of gestational age variation on baseline behaviours more explicitly.
-
Justify the dichotomization of ethnicity and potential implications for interpreting associations with BMI.
5. Results
The results are detailed and well organized, with effective use of tables to display complex site-level comparisons. The multivariate analysis adds valuable depth, particularly in identifying factors associated with BMI within each site.
Strengths:
-
Rich cross-sectional dataset with appropriate multivariable modelling.
-
Insightful findings showing the variability in how demographic and behavioural variables relate to BMI.
Suggestions:
-
Emphasize more clearly which variables consistently correlate with BMI across multiple sites (e.g., education, health literacy).
-
Consider visualizing key site comparisons (e.g., BMI, PA, energy intake) with figures to enhance readability.
6. Discussion
The discussion thoughtfully interprets the findings, aligning them with national trends in obesity, GDM screening practices, and known sociocultural contexts. The limitations are candidly acknowledged.
Strengths:
-
Contextualization of BMI differences concerning national prevalence data.
-
Consideration of structural determinants like education and housing.
Areas for Expansion:
-
The implications for tailoring GDM interventions in multicultural or migrant populations could be further explored.
-
Include brief commentary on how findings might inform updates or refinements to GDM risk screening tools in diverse healthcare settings.
7. Conclusion
The conclusion appropriately summarizes key findings and their practical implications. The call for tailored prevention strategies and validation of risk assessment tools across contexts is well supported.
Suggestion:
-
Add a brief statement on the need for follow-up studies (e.g., longitudinal tracking of GDM onset or maternal-infant outcomes in this cohort).
8. Ethical Considerations and Reporting
All ethical approvals are documented. Data sharing and funding statements are complete. There are no declared conflicts of interest, and authorship contributions are transparent.
9. Language and Structure
The manuscript is well-written, clear, and professional throughout. Terminology is appropriate, and the tone aligns with scientific publishing standards.
Suggestion:
-
A brief language edit to eliminate minor redundancies (e.g., repetitive mentions of site names when already clarified in tables).
Final Evaluation
This study provides a valuable contribution to the literature on GDM prevention and maternal health disparities. With minor revisions to enhance clarity, contextual interpretation, and data presentation, it will make a strong publication in IJERPH.
Author Response

(The authors gave the same response as above.)

Reviewer 4 Report
Comments and Suggestions for Authors
-There are multiple grammatical errors and overuse of parentheses.
-The authors have used “like” multiple times to make examples. Like is a polysemy, so replace it throughout the manuscript.
-For 4 international sites, the authors used both country names and city names throughout the manuscript, which made it confusing; country names in some places and city names in other places. It would be better to use either country names or city names throughout the manuscript. For example, in Tables, 4 country names are listed in the table title, while the city names are column titles. In addition, a listing of country names or city names should be in the same order throughout the manuscript.
-Power analysis: add sample size calculations conducted by power analysis.
-Measures: briefly add what demographic and clinical characteristics were collected.
-Measures: the sentence, “height, weight, and blood pressure…; self-reported measurements were used..”. Multiple studies clearly reported inaccuracies of self-reported height and weight data, leading to underestimation of BMI and misclassification of obesity. Examining factors associated with BMI is one of the study objectives. Therefore, self-reported height and weight data should not be used in data analysis.
-Data analysis: 1) There is no explanation of the Welch’s F test. 2) Add what specific post-tests were used for post-hoc analyses. 3) The data analysis section should be rewritten in simple, short sentences for the readers’ understanding.
-Figure 1: For 312 individuals not invited to participate, why were they not invited to participate?
-Tables: 1) In the “total” column, there is n=804 in a parenthesis. The information (n = 804) is incorrect because some data, such as parity in Table 2, includes 797, not 804. Take out n=804 in a parenthesis. 2) For post-hoc analyses, indicate significant differences among groups. 3) All abbreviations and acronyms used in tables should be defined in the table note. 4) consistent order of cities and counties throughout tables and manuscript.
-Table 2: Check GDM risk score data. One of the eligibility criteria is pregnant women scoring the same as or greater than 3 on the GDM screening tool. All means and SDs of the GDM risk score do not seem to support this criterion.
-Clinical characteristics: It appears that only systolic and diastolic blood pressure measurements were recorded, and these were classified as clinical characteristics. In that case, refer to them as blood pressure, not clinical characteristics, throughout the manuscript.
-Cultural patterns in health behaviors: Take out “cultural patterns” throughout the manuscript because there is no content about true cultural patterns.
-Take out p>0.05 in text throughout the manuscript.
-Table 3: were intakes of fat, protein, and carbohydrates adjusted by total energy intake? They should be adjusted by total energy intake; simply comparing the macronutrient intakes is not meaningful.
-Tables 4-6: 1) Regarding “BMI-D model including dietary variables”, what specific dietary variables were included in the model? The info should be added. It seems that only total dietary energy was included in the model. In that case, it should be BMI-energy, not BMI-D. 2) The tables should be re-created to enhance readers’ understanding and to compare the data among 4 sites easily and immediately.
-Tables 5 & 7 BMI-D: The total energy intake of participants in 4 cities was reported in Table 3. The number of participants' energy intake in Table 3 does not match the number of participants’ data used in BMI-D in Tables 5 and 7. In Table notes, It’s said “BMI-D model excluding implausible energy reporters”. All implausible energy reporters must also be excluded from all data analysis.
Discussion lines 244-246: Given the information, participants in the Granada cohort cannot be representative of Spain. How was each site per country chosen?
-There is no “cultural” related aspect in the manuscript, so take out “cultural” throughout the manuscript.
Comments on the Quality of English Language
The English could be improved to more clearly express the research. There are multiple grammatical errors and overuse of parentheses. In addition, multiple sentences are too long.
Author Response

(The authors gave the same response as above.)
